# Estimating Future Residential Property Risk Associated with Wildfires in Louisiana, U.S.A.

**Rubayet Bin Mostafiz** [1,*] **, Carol J. Friedland** [2] **, Robert V. Rohli** [1,3] **and Nazla Bushra** [1]

1   Department of Oceanography & Coastal Sciences, College of the Coast & Environment, Louisiana State University, Baton Rouge, LA 70803, USA; rohli@lsu.edu (R.V.R.); nbushr1@lsu.edu (N.B.)
2   LaHouse Home and Landscape Resource Center, Louisiana State University Agricultural Center, Baton Rouge, LA 70803, USA; cfriedland@agcenter.lsu.edu
3   Coastal Studies Institute, Louisiana State University, Baton Rouge, LA 70803, USA
*   Correspondence: rbinmo1@lsu.edu

**Abstract:** Wildfire is an important but understudied natural hazard in some areas. This research examined historical and future wildfire property risk at the census-block level in Louisiana, a U.S.A. state with relatively dense population and substantial vulnerability to loss from wildfire, despite its wet climate. Here wildfire risk is defined as the product of exposure and vulnerability to the hazard, where exposure is a function of the historical and anticipated future wildfire frequency/extent, and vulnerability is a function of population, structure and content property value, damage probability, and percent of properties damaged. The results revealed a historical (1992–2015) average annual statewide property loss due to wildfire of almost USD 5.6 million (in 2010 USD), with the greatest risk in southwestern inland, east-central, extreme northwestern, and coastal southwestern Louisiana. The geographic distribution of wildfire risk by 2050 will remain similar to that today, but the magnitude of losses was projected to increase statewide to over USD 11 million by 2050 (in 2010 USD), an increase of more than 100% over 2010 values. These estimates are conservative, as they did not include crop, forestry, or indirect losses (e.g., cost of evacuation and missed time at work). Overall, results suggested that increased efforts are needed to contain wildfires, to reduce the future risk of this increasing and underestimated hazard.

**Keywords:** wildfire; natural hazards; population projections; forest resources; vulnerability; resilience; environmental change; climate change; burn probability

## 1. Introduction

Although weather-related disasters cause extensive and rapidly increasing damage worldwide, efforts to understand the holistic risk from these hazards are still in progress. While a growing amount of research is focusing on assessing risk due to floods [1–5], hurricanes [6], tornadoes [7], and extreme weather events [8], the risk of wildfire—combustion in a natural setting, marked by flames or intense heat, ranging in coverage from less than 20 hectares to over 3 million hectares—is lesser-studied. Natural and human-prescribed fire is often healthy [9,10] as a mechanism for restoring nutrients to the soil and providing new niches while often leaving native species unharmed or resilient to the disturbance [11], for combatting pests, diseases, and fungal growth, and for allowing for the post-fire regrowth to establish hardier individuals. However, the hazardous aspects of wildfires deserve more attention in risk assessments.

The risk of wildfire is particularly important to understand near the wildland-urban interface (WUI; [12])—the area where development meets wildland vegetation, with both providing fuel for fires, leaving more natural ecosystems, people, and property exposed to wildfire danger [13], and in locations that have previously been understudied. Because wildfire is a critical ecosystem process influenced by a combination of natural and

human factors [14] and because its presence and intensity can be modified by climate change [15–22], it must be considered in environmental risk assessment.

The impacts of wildfire are three-fold: environmental, health, and property. In recent decades, research on environmental impacts has emphasized wildfire dynamism and variability at the regional level [16,23–36], in part because climate change implications on ecosystems [37] present unique challenges for hazard management in each wildfire regime [38]. Regarding the human health impacts of wildfire, the Southeastern U.S.A. has been found to be most affected by hospital admissions and premature deaths due to wildfire events in the U.S.A. [39], in addition to generally unhealthy conditions due to wildfire-related smoke [40,41].

This paper presents a census-block-level property risk assessment for wildfire in Louisiana, U.S.A., in contrast to most of the existing wildfire research in the U.S.A., which focuses on the Western U.S.A. [18,42–45]. The three primary objectives were to (1) characterize the historical wildfire burn probability, (2) project the future wildfire burn probability, and (3) assess the future property loss in Louisiana due to wildfire. The integration of natural and social science approaches as shown in the three objectives here was needed to understand more fully the property risk of wildfire, especially in light of climate change concerns [46]. The contribution of this paper is to provide a more complete understanding of the wildfire risk, at the census block level, by extending previous innovative geospatial approaches to overlaying fire extent with infrastructure [47] quantitatively and by incorporating changes in population, property value, and climate, along with the historical wildfire burn probability, damage probability, and percent of damaged property as components of the property risk. The results will benefit foresters, property owners, and mitigation specialists within and beyond Louisiana as they seek new and improved ways to characterize and prepare for the wildfire hazard.

## 2. Background: Wildfire-Related Property Impacts

Many recent studies point to the substantial property loss associated with wildfire, especially at the WUI. Property damage in northeastern Florida due to wildfires in the El Niño year of 1998 was estimated at USD 10–12 million [48]. In a holistic cost–benefit or "hedonic" approach, Ref. [49] found a negative effect of wildfire on property values in California, Colorado, and Montana, with less conclusive evidence from research based elsewhere in Colorado and Alaska. Further work showed that home prices and sales rates in the Front Range of Colorado are influenced by wildfire risk and risk perceptions [50]. Other research has examined the impacts of wildfire risk on residential property values in the Netherlands [51]. More recently, programs like FireSmart [52] provide homeowners in the WUI with information to make more informed decisions for protecting their property from the wildfire hazard.

Other studies on modeling property risk due to wildfire have emphasized changing populations and mobility, especially as they interface with weather variables. For example, Ref. [53] used an artificial neural network approach to model the impact of population density and weather parameters, such as average relative humidity, wind velocity, and daily sunshine hours, on forest fire risk in Japan. Ref. [54] assessed the exposure of resources to wildfire in light of population patterns. Incorporating both natural and anthropogenic ignitions, Ref. [42] proposed a wildfire simulation model that characterized potential wildfire behavior in terms of annual burn probability and flame length in the Oregon and Washington national forests. Ref. [55] examined the spatiotemporal patterns of the wildfire occurrence in Sardinia, Italy, and characterized the outcomes of both the probability of ignition and large fire in terms of weather, land use, anthropogenic features, and time of year. Based on wildfire likelihood and intensity over the 1992 to 2010 period, Ref. [56] developed a broad-scale wildfire potential map for the contiguous U.S.A. that can be used to analyze wildfire threat or risk to structures or power lines. Ref. [57] developed a wildfire prediction model incorporating ten geophysical and climatological parameters

to investigate the spatial distributions of wildfire probabilities from 32 fire events at the Zagros ecoregion of Iran.

The most recent research has emphasized more sophisticated incorporation of uncertainty in wildfire property risk analyses. Based on the minimum travel time algorithm, Ref. [58] developed a fire simulation model to analyze the wildfire exposure of highly valued resources and assets in a 28,000 ha area in central Navarra, Spain. Ref. [59] generated a GIS-based novel hybrid artificial intelligence approach to model spatial susceptibility of the wildfire hazard in the central highland forest region of Vietnam. Ref. [60] built a probabilistic model for predicting wildfire housing loss at the mesoscale (1 km$^2$) level using Bayesian network analysis, enabling the construction of an integrated model based on causal relationships between the influencing parameters jointly with the associated uncertainties. Ref. [61] predicted the spatial pattern of wildfire susceptibility in Huichang County, China, by using the integrated probabilistic "weights-of-evidence" and knowledge-based "analytical hierarchy process" models. In recent years, the use of rigorous data interpretation techniques such as geographical information systems (GIS), and statistical and machine learning approaches have resulted in various prediction models of wildfire probability [57,59,61–70].

The U.S. Forest Service Missoula Fire Sciences Laboratory's geospatial Fire Simulation system (FSim) has become increasingly useful for property risk assessments in the most recent years. Ref. [71] used FSim to create national burn probability and conditional fire intensity level estimated at a 270 m grid resolution over the contiguous United States. FSim includes scenarios for generating wildfire-conducive weather (including lack of humidity in areas of combustible vegetation cover), wildfire occurrence, fire growth, and fire suppression.

## 3. Temporal Trends in Wildfire Occurrence in the U.S.A.

The historical record shows that the Western U.S.A. has generally experienced increasing wildfire frequency and intensity over time [43,44,72–76]. Despite earlier research that suggested that the Mississippi wildfire occurrence had decreased since the 1920s [77], wildfire risk-related research on the Southeastern U.S.A. deserves more attention. This is because of the droughts in recent years [78], dense population, and high probability of risk from other wildfire-related hazards. Louisiana is particularly understudied regarding wildfire, especially in light of the suggestion of [39] that PM2.5 concentrations in Louisiana attributed to wildfire exceeded that of any other state except California in 2008.

## 4. Study Area

Louisiana was selected as the focus of this research for several reasons. Despite abundant rainfall, the state can be subjected to periods of drought [79] and therefore wildfire, which can have disproportionate impacts because of the heavy reliance of wet-environment land uses, such as rice farming, industrial applications, and recreation. Moreover, as is shown later, the future frequency of such periods of wildfire is expected to increase in Louisiana as in much of the rest of the United States. Furthermore, the state is relatively densely populated compared to most of the wildfire-vulnerable Western U.S.A., causing anthropogenic activity to contribute substantially to the intensifying hazard and its human impacts. Finally, Louisiana-based studies on wildfires to date are largely limited to environmental impacts [80–87], with substantially less work on risk assessment regarding property.

Over the 2007 to 2016 period, an annual average of 1431 wildfires burned 14,950 acres of forestland in Louisiana, with most of these fires caused by arson or human negligence, exacerbated by human confrontation with nature [88]. Likewise, lightning was found to be a minor cause of wildfire in nearby Mississippi compared to anthropogenic causes [77]. Using the customary categorization of U.S. wildfires as large (>300 acres) or small (<300 acres), Louisiana wildfires tend to be small, averaging about 10 acres in size [88]. Wildfire as tabulated here does not include prescribed burns [89] or fires started in a building.

## 5. Materials and Methods

### 5.1. Data

Because wildfire outside but near Louisiana can endanger the state, a 50-km buffer to include the adjacent Texas, Arkansas, and Mississippi was analyzed along with Louisiana. To characterize the historical wildfire probability, historical wildfire occurrence data from 1992 to 2015 from [90] and large wildfire burn probability from [91] were used. These consider vegetation and other types of land cover. Projecting the future wildfire probability relies on information from the fourth National Climate Assessment [92]. Ref. [92] follows the method of the Intergovernmental Panel on Climate Change (IPCC) by running fossil fuel emission scenarios termed "representative concentration pathways" (RCPs), with the scenarios numbered based on the amount of radiative forcing (in W m$^{-2}$) anticipated in the year 2100, such that RCP8.5 is the most severe scenario. As in the vast majority of contemporary climate change-based research, the model results using the RCPs are based on the Coupled Model Intercomparison Project (CMIP; [93]). Results from IPCC's fifth assessment report are available in [92]. Other scenarios account for changes in economic growth, environmental values, globalization, and regionalization. Louisiana census-block shapefiles were downloaded from [94], and population projections are based on data from [95]. Louisiana Department of Agriculture and Forestry (LDAF) detailed fire summary data for Louisiana (2007–2017; [96]) serve as a baseline for future property loss due to wildfire.

### 5.2. Assessing Historical Wildfire Burn Probability

A method of computing both large and small fire probabilities is necessary here because FSim focuses on large (>300 acres) fires [56] and fires in and near Louisiana are primarily classified as "small." The wildfire probability calculation follows the method of [56], which uses the large fire (i.e., >300 acres) probability from FSim [97] supplemented by a collection of small fire (i.e., <300 acres) probabilities from the Fire Protection Agency (FPA) fire occurrence data [98]. The total probability was calculated as the sum of large and small fire probabilities.

Large wildfire burn probability raster files were downloaded from the U.S. Department of Agriculture [91]. Then, the large fire probabilities in Louisiana and its surroundings (50 km buffer) were extracted. To extract the small fire probabilities, the nationwide fire occurrence point-based shapefiles (1992–2015) from the U.S. Department of Agriculture [90] were downloaded and fires larger than 300 acres were removed using GIS applications. The small wildfire occurrence were then extracted for Louisiana and its surroundings. A total of 73,501 small fire records existed in the study area. Planar kernel density analysis [99] was then performed for the small fire data to produce a spatial distribution of fire density, with a cell size set to 270 m to correspond to that used in the FSim layer, and a kernel size of 50 km [56]. The cell area (270 m × 270 m = 72,900 m$^2$) was then multiplied by the resulting kernel density of fire. To identify the small fire probability over the smoothed surface, the total number of fires (73,501) was divided by the kernel density, and this value for each pixel was divided by 24 to compute the historical (1992–2015) annual probability. To calculate the total fire annual probability, this small fire probability was added to the large fire probability from FSim. The 50 km buffer was then removed via masking with the Louisiana boundary. Finally, the wildfire burn probability of each census block $p(f)_i$. was extracted from the raster files at the centroid of each census block.

### 5.3. Assessing Future Wildfire Burn Probability

The first step in determining future wildfire burn probability was to quantify the wildfire hazard for Louisiana. Ref. [100] modeled seasonal changes using the Keetch–Byram Drought Index (KBDI; [101]), which has been commonly used to assess wildfire probability [102], at the global scale. The A2a economic or environmental or globalization or regionalization scenario, which assumes that the global population surpasses 10 billion by 2050, with relatively slow economic and technological development, was found to create

global $CO_2$ mixing ratios of 575 parts per million (ppm) by 2050 and 870 ppm by 2100 [100], compared to the current 418 ppm. Thus, it is not surprising that models consistently project a warming global atmosphere [103], which would seemingly increase wildfire probability.

Nevertheless, such a projection of the future wildfire hazard intensity is not straightforward. Anticipated water scarcity and intensifying insect infestations, such as by the southern pine beetle in Louisiana [104] in a warming world may mitigate the wildfire hazard by reducing fuel from trees that are stunted or killed [103,105]. Other factors that might at first glance seem to mitigate future wildfire occurrence may actually exacerbate the wildfire hazard. For example, although daily precipitation totals are projected to increase by 9–13% for Louisiana by 2050 amid a generally more extreme precipitation climate nationwide by 2100 [103], the enhanced "per event" precipitation and the sharp increase in the frequency of days having a greater than 90th percentile of precipitation are accompanied by substantially more frequent "zero precipitation days" and small precipitation totals that would fall within today's zero-to-tenth-percentile [103]. This is because the temporally warming atmosphere would require more atmospheric moisture before saturation is reached, and therefore before precipitation could occur. Thus, the anticipated increased precipitation totals could result in an enhanced wildfire hazard for Louisiana. Furthermore, the anticipated weakening of steering circulation [106] that moves frontal and tropical weather systems will leave longer interarrival times between intense precipitation events. Such changes in both precipitation intensity and interarrival times would reduce soil moisture, which in turn would increase the wildfire burn probability.

Ref. [103] acknowledged that projections of seasonal precipitation deficits lack confidence, particularly regarding extratropical precipitation extremes [107] and that resulting wildfire occurrence is likely to display great local and regional spatial heterogeneity. The interplay between modeled trends in individual variables that may have compensating effects in their influence on future wildfire intensity and probability also complicates regional generalizations. Nevertheless, Ref. [103] recognized that the preponderance of evidence suggests that enhanced evapotranspiration caused by increased temperatures will outpace the projected increasing precipitation totals. The net result is likely to be soil desiccation through this century over much of the continental U.S.A., at least under the RCP8.5 scenario.

At the regional scale, Ref. [105] used three general circulation models and three IPCC-based emission scenarios to conclude that median annual area of the Southeastern U.S.A. affected by lightning-ignited wildfire will increase by 34%, human-ignited wildfires will decrease by 6%, and total wildfire will increase by 4% by 2056–2060 compared with the years 2016–2020. Such results are corroborated by [92], which suggestd an increase in lightning-ignited wildfire by 2050 in the Southeastern U.S.A., including Louisiana [103].

The [103] projection for Louisiana is for small soil moisture decreases in autumn relative to natural variability but large decreases relative to natural variability in the other three seasons by mid-century, with a "medium" degree of confidence [103]. The earlier KBDI work [100] (their Figure 5) estimated a similar result for Louisiana, but with more precise soil moisture forecasted decreases of 50–150 mm per three-month period in autumn and winter (September through February) and decreases of 200–250 mm per three-month period in March through May and June through August. The midpoint of the time series of the [100] projection was 2085; therefore our current research assumed that half of the projected changes will occur by 2050. Thus, decreases of 25–75 mm per three-month period (or 8–25 mm per month, with 17 mm per month as the midpoint) were projected for each month from September through February in Louisiana by 2050. Decreases of 100–125 mm per three-month period (or 33–42 mm per month, with 38 mm per month as the midpoint) were projected for each month from March through August in Louisiana by 2050.

To provide more detail for Louisiana based on these results from [100], the mean monthly precipitation data for 31° N, 91.5° W (the nearest available data point to the center of the state) were input into the Web-based, Water-Budget, Interactive, Modeling Program [108,109]. WebWIMP calculates decreases in soil moisture in the upper layers

of 12.2% (February) to 46.1% (August). Thus, a 25% decrease in available moisture in the organic matter and uppermost soil layers, and a 25% increase in wildfire susceptibility across Louisiana by 2050 ($F_{2050}$ = 1.25) was projected here. These calculations are not without their caveats. For example, these changes did not take into account projected changes in global air temperature. According to [110], historically unprecedented warming by 2100 is to be expected in Louisiana under a higher emissions pathway.

*5.4. Projecting Population*

The method of projecting population (*P*) at the census-block (*i*) scale by the year 2050 followed that of [111–114]. Because annual census-block level population estimates are unavailable, the process begain with parish- (i.e., county-) wide annual growth rate calculations. For each of Louisiana's 64 parishes (*j*), the average annual population growth rate ($r_j$) for the *n*-year (i.e., 40) period for which annual estimates [95] exist (i.e., 1980–2020) was calculated, beginning in year *y*, as described by Equation (1):

$$r_j = \frac{\sum_y^{y+n} \left[ \frac{(P_{j,y+1} - P_{j,y})}{P_{j,y}} \right]}{n} \tag{1}$$

After $r_j$ was determined for each of Louisiana's 64 parishes, the future population change was downscaled to the census block (*i*), assuming that $r_j$ was the same for each census block in its parish. The future population is then projected for each census block, assuming that the currently unpopulated census blocks remained uninhabited through to 2050. For each *i*, the 2010 population was used as the base (i.e., $P_{0,i} = P_{2010,i}$) and the future population was projected out to 2050 (i.e., $P_{f,i} = P_{2050,i}$), given an n-year period within which the population changes (*t*), as shown in Equation (2):

$$P_{f,i} = P_{0,i} e^{r_j t} \tag{2}$$

This approach outperformed other methods that were tested. Specifically, the extension of a trend line of the parish-level population into the future proved impractical because several parishes showed an insignificant trend line and low explained variance. A second methodology tested was the extension of the growth rate trend line to approximate the 2050 population, but this proved problematic for the same cause. The abrupt, sizeable, and temporary population redistributions both within and beyond Louisiana in the wake of significant hurricanes (most notably Katrina in 2005), were likely contributors to the low explained variance. The procedure selected was least sensitive to these concerns and was also implemented effectively in [111,113,114].

*5.5. Assessing Structure and Content Value*

To evaluate the current and future structure values (*SVs*) in each census block, the total number of buildings in each census block in 2010 ($N_{2010,i}$) was acquired from [94] by summing the buildings constructed during each time interval as reported in the shapefiles [112]. Then, this value was multiplied by the mean building value in 2010 in a given census block ($MV_{2010,i}$) to give the total *SV* in that census block ($SV_{2010,i}$), as shown in Equation (3):

$$SV_{2010,i} = N_{2010,i} \times MV_{2010,i} \tag{3}$$

The number of buildings in 2050 in a census block ($N_{2050,i}$) was assumed to change proportionately to population; therefore, the population projection described above was used to scale the building inventory. The total *SV* in 2050 in a census block ($SV_{2050,i}$) was then calculated as the product of $SV_{2010,i}$ and the ratio of 2050 population to 2010 population (Equation (4)).

$$SV_{2050,i} = SV_{2010,i} \times \frac{P_{2050,i}}{P_{2010,i}} \tag{4}$$

The economic but not sentimental worth of items on the damaged property is known as the content value ($CV$; [114]). Ref. [115] calculated $CV$ by multiplying $SV$ by a structure-to-content value ratio according to the type of occupancy in its National Structure Inventory version 2.0 (i.e., NSI 2.0). $CV$ was presumed to be equal to $SV$ for residential, commercial, and industrial structures in NSI 2.0 [115]. Ref. [115] recommended that $CV$ be assumed to be 50% of $SV$ when its official depth-damage functions are unavailable, and [116] (pp. 6–9) suggested assuming that $CV$ is half of $SV$ for all residential structures. As this research included only residential structures, here the midpoint of these two estimates was taken (Equation (5)).

$$CV = 0.75 \, SV \tag{5}$$

The property value ($PV$) is the sum of $SV$ and $CV$ (Equation (6)).

$$PV = SV + CV \tag{6}$$

*5.6. Projecting Future Property Loss*

Because the LDAF records show that from 2007 to 2017, 389 of the 12,979 Louisiana residences that were threatened by fire were damaged [96], a conditional probability of damage $p(d|f_i)$ of 0.03 was assumed (see Supplementary Materials). Then, the probability of damage $p(d)_i$ was calculated as shown in Equation (7):

$$p(d)_i = p(f)_i \times p(d|f_i) \tag{7}$$

Based on LDAF advice, each damaged building was assumed to have a loss of 5% of the $PV$ [96]; thus, $d$ was 0.05. Future property loss due to wildfire ($L$) in census block $i$ was calculated as shown in Equation (8):

$$L_{2050,i} = PV_{2050,\,i} \times p(d)_i \times d \times F_{2050,i} \tag{8}$$

All losses are expressed in 2010 USD. Because of uncertainties in these assumptions, a sensitivity analysis was conducted to demonstrate the impact of model assumptions regarding $F_{2050}$, $SV$ to $CV$ ratio, $p(d|f_i)$, and $d$. More specifically, each of these assumed values was changed by +50%, to give a range of values for $L_{2050,i}$.

Note that the calculation for historical annual property loss ($\overline{L}_{1992-2015,i}$) is simply

$$\overline{L}_{1992-2015,i} = PV_{2010,\,i} \times p(d)_i \times d \tag{9}$$

with the 2010 property value being used at each census block $PV_{2010,i}$. Annual per capita and per building property loss in 2010 and 2050 by census block were calculated by dividing by the population and building count, respectively.

To evaluate the current and future structure value ($SV$) in each census block, the total number of buildings in each census block in 2010 ($N_{2010,i}$) was acquired from [94] by summing the buildings constructed during each time interval as reported in the shapefiles [113]. Then this value was multiplied by the mean building value in 2010 in a given census block ($MV_{2010,\,i}$) to give the total $SV$ in that census block ($SV_{2010,i}$), as shown in Equation (3).

## 6. Results
*6.1. Historical Wildfire Probability*

The frequency of historical (1992–2015) wildfire incidents in the top twelve parishes is shown in Figure 1A. The historical (1992–2015) wildfire burn probability ranged from 0 in coastal southeastern census blocks to 7.7% at a point in Cameron Parish (Appendix A), in the extreme coastal southwest (Figure 1A). The west-central, east-central, and extreme northwestern and southwestern parts of the state had the highest burn probability for wildfire (Figure 1A). For parish-level planning purposes, it is also worthwhile to note that Washington (the northeasternmost parish north of New Orleans and Lake Pontchartrain)

was the most vulnerable parish, where the mean historical wildfire burn probability was 4.1% (Appendix A).

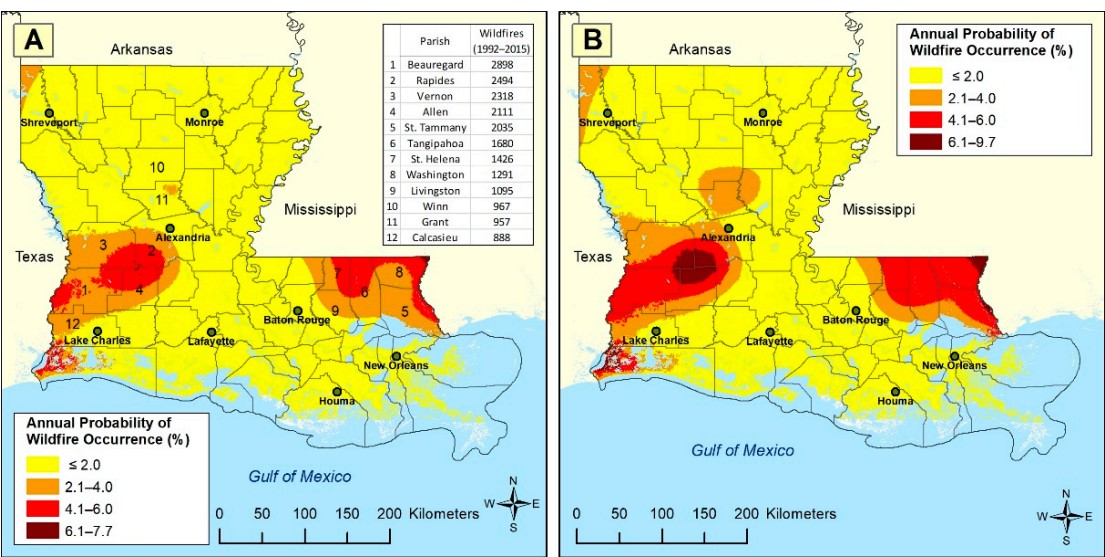

**Figure 1.** Wildfire burn probability: (**A**) historical (1992–2015), with the top 12 parishes for historical wildfire frequency labeled, and (**B**) projection for 2050.

### 6.2. Projected Future Wildfire Probability

The projected wildfire burn probability for 2050 is anticipated to range from 0 on the southeastern coast to 9.6% at a point in Cameron Parish (Figure 1B; Appendix B). The highest wildfire burn probability among census block centroids is expected to be 8.6% in census block 221179501012000 in Washington Parish. Washington will be the most vulnerable parish, where the mean projected wildfire burn probability is 5.2% (Appendix B). Washington, St. Helena, Beauregard, Allen, Tangipahoa, St. Tammany, Vernon, Rapides, Livingston, and Calcasieu are the top ten most vulnerable parishes in Louisiana (Appendices A and B), whereas St. Mary, Iberia, Terrebonne, Assumption, and Lafourche are the least vulnerable parishes (Appendices A and B). In general, the wildfire hazard is likely to remain concentrated in the same geographical areas of the state as in the historical record, but burn probabilities are likely to increase (Figure 1A,B).

### 6.3. Projected Future Population

Using the values calculated in Equations (1) and (2), and assuming that the 102,781 census blocks in Louisiana that were inhabited in 2010 (from among the 203,447 total) will remain the only blocks inhabited in 2050, the 2050 population density projection was generated. The population is the greatest around New Orleans, Baton Rouge, and Shreveport, the state's three largest cities (Figure 2A). By 2050, the population will remain concentrated in largely the same areas, but with increased population especially around Lafayette, Baton Rouge, and north of New Orleans and Lake Pontchartrain (Figure 2B). Population decreases are expected throughout northeastern Louisiana, along the Red River Valley from Shreveport to the area southeast of Alexandria, in the New Orleans area, and elsewhere (Figure 2B). Appendix C shows these values by parish.

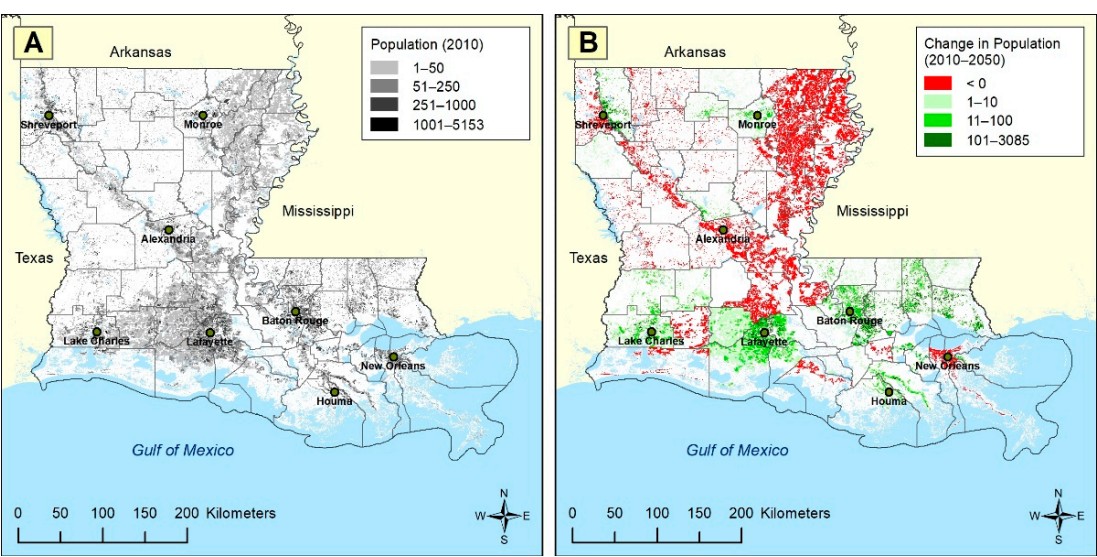

**Figure 2.** Population by census block: (**A**) 2010, and (**B**) change in population from 2010 to 2050.

*6.4. Historical and Projected Future Property Loss*

The historical (1992–2015) average annual statewide property loss due to wildfire was USD 5,556,389 (in 2010 USD). The wildfire risk is projected to increase statewide by 2050 due to intensifying negative impacts of climate change statewide, and increasing population and development in many parts of the state, with projected annual loss of USD 11,167,496 by 2050 (in 2010 USD; Appendix D), an increase of 101%. Note that these values do not include crop, forestry, or indirect losses (cost of evacuation, missed time at work, etc.), which are likely to be high for wildfire as well. Thus, the loss estimates here are conservative.

The maximum estimated property losses will remain concentrated near their present locations, namely east-central, southwestern, and northwestern Louisiana (Figure 3A,B). On a per capita basis, the historical (1992–2015) average annual per capita property loss due to wildfire was only USD 1.23 (in 2010 USD) in Louisiana (Appendix D). Projected per capita property loss is USD 1.97 by 2050 (in 2010 USD), giving an increase in annual per capita property loss of 61% (Appendix D). The same general spatial distribution of per capita property losses (Figure 4A,B) occured (and is projected to occur by 2050), as was shown for absolute losses. The historical average annual per building property loss was USD 2.83 (in 2010 USD) whereas the projected loss will be USD 4.63 (in 2010 USD) by 2050 (Appendix D). Thus, the annual per building property loss is projected to increase by 64% in Louisiana. Again, the spatial distribution remains similar (Figure 5A,B).

At the parish level, St. Tammany (immediately north of Lake Pontchartrain and New Orleans in east-central Louisiana) had the highest historical (1992–2015) overall wildfire annual property loss (USD 1,560,580), per capita property loss (USD 6.68), and per building property loss (USD 16.36) among the parishes (Appendix D). Changes in the wildfire burn probability and expansion of population are projected to change the wildfire risk by 2050. Nevertheless, the greatest annual wildfire property loss (USD 4,633,439), per capita property loss (USD 8.34), and per building property loss (USD 20.45) are expected to remain in St. Tammany Parish (Appendix D).

At the census-block level, the highest historical average annual property loss due to wildfire was in block 221030408035041 of St. Tammany Parish (USD 18,837). The highest historical (1992–2015) average annual per building property loss (USD 37.35) was in census block 221030407061032, also in St. Tammany Parish. The highest historical annual per capita property loss in the state was USD 354.83 in census block 220919511002011, in St. Helena Parish (east-central Louisiana, northwest of St. Tammany).

By 2050, the greatest annual property loss due to wildfire is projected to be in census block 221030408035041, in St. Tammany Parish (USD 55,950). The highest annual per capita property loss (USD 443.54) is projected to be in census block 220919511002011 of St. Helena Parish. The highest annual per building property loss (USD 46.68) is projected to be in census block 221030407061032, in St. Tammany Parish.

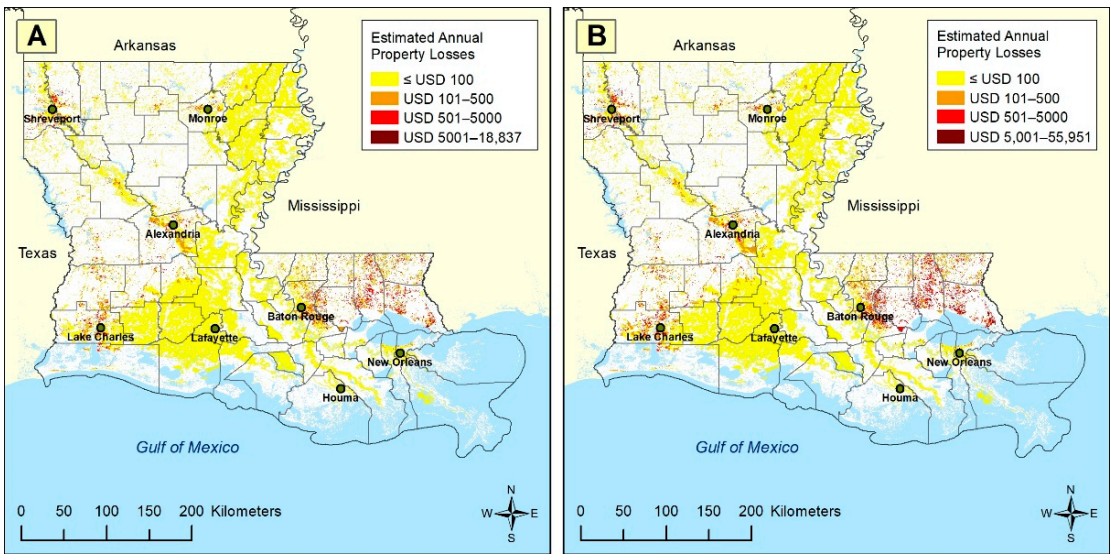

**Figure 3.** Estimated annual property loss (2010 USD) due to wildfire by census block: (**A**) historical (1992–2015) and (**B**) 2050.

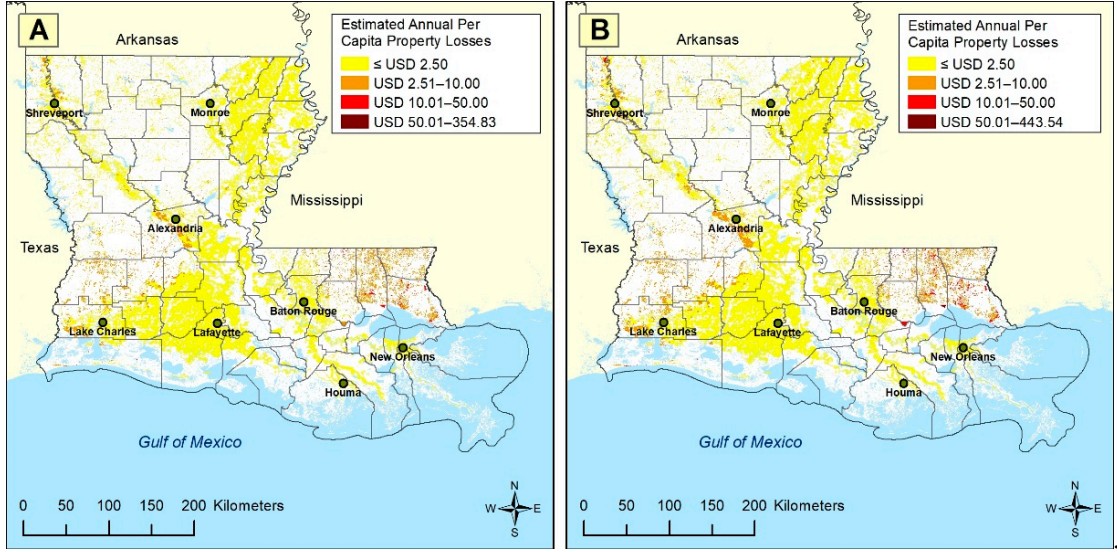

**Figure 4.** Estimated annual per capita property loss (in 2010 USD) due to wildfire by census block: (**A**) historical (1992–2015), and (**B**) 2050.

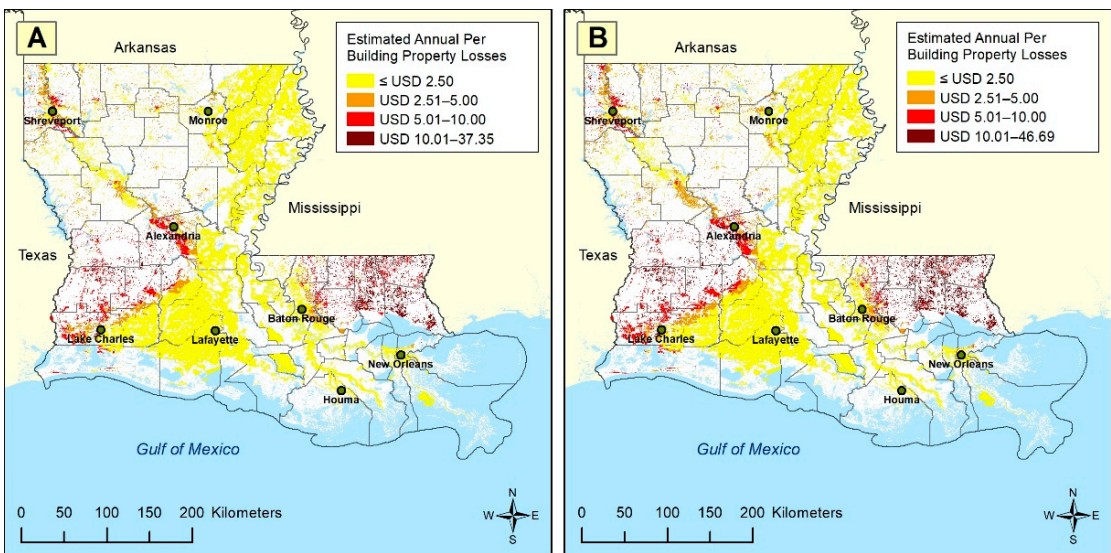

**Figure 5.** Estimated annual per building property loss (in 2010 USD) due to wildfire by census block: (**A**) historical (1992–2015), and (**B**) 2050.

*6.5. Sensitivity Analysis*

If the wildfire burn probability in 2050 increases by 12.5% (i.e., $F_{2050} = 1.125$) or 37.5% from the present (i.e., $F_{2050} = 1.375$), rather than the 25% increase (i.e., $F_{2050} = 1.25$) currently assumed, the result would change by +10.0% (Table 1). If $CV$ would be taken to be only 37.5% of $SV$. or 112.5% of $SV$, instead of the model-estimated 75% of $SV$, the sensitivity was approximately +21.4% (Table 1). However, the model was much more sensitive to variations in the other two variables included. Specifically, the greatest sensitivity in model assumptions was that for $p(d|f_i)$, which was derived as the conditional probability of damage (0.03). If $p(d|f_i)$ was actually 0.015 or 0.045, the modeled annual loss in 2050 would change by 50% (Table 1). Same as sensitive was the percent of property damage (*d*). If 2.5 or 7.5% of the property is damaged, rather than the 0.05 assumed in the model, the annual loss in 2050 would change by 50% (Table 1).

**Table 1.** Sensitivity analysis of 2050 projections of Louisiana statewide annual property loss (i.e., risk) due to wildfire, by parameter (in 2010 USD).

| Parameter | Low Scenario | Modeled (Equation (6)) | High Scenario | Difference from Equation (8) (%) |
|---|---|---|---|---|
| Future Condition ($F_{2050}$) | USD 10,050,746 (+12.5%) | USD 11,167,496 (+25%) | USD 12,284,245 (+37.5%) | ±10.0 |
| Content to Structure Value Ratio ($CV/SV$) | USD 8,774,461 $CV = 0.375\,SV$ | USD 11,167,496 $CV = 0.75\,SV$ | USD 13,560,530 $CV = 1.125\,SV$ | ±21.4 |
| Conditional Probability of Damage $p(d|f_i)$ | USD 5,583,748 ($p(d|f_i) = 0.015$) | USD 11,167,496 ($p(d|f_i) = 0.03$) | USD 16,751,244 ($p(d|f_i) = 0.045$) | ±50.0 |
| Percent of Property Damage (*d*) | USD 5,583,748 (*d* = 0.025) | USD 11,167,496 (*d* = 0.05) | USD 16,751,244 (*d* = 0.075) | ±50.0 |

## 7. Discussion

While it is tempting to overlook the wildfire hazard in a state that receives abundant rainfall, is susceptible to so many other, more calamitous hazards, and often suffers from other more pressing economic hardships, the wildfire hazard in Louisiana is formidable

and is expected to become more challenging in coming decades. The mean annual loss (1992–2015) of USD 5.6 million (in 2010 USD) is projected to increase to USD 11.2 million (in 2010 USD) statewide by 2050. Wildfire is projected to be more costly than several other hazards in the state, including lightning (USD 4.3 million; [111]), hail (USD 2.5 million; [111]), and sinkholes (USD 0.3 million; [114]). Nevertheless, costlier hazards in Louisiana, such as expansive soils (USD 91.8 million; [113]), tornadoes (USD 24.3 million; [111]), extreme cold (USD 23.2 million; [111]), and wind (USD 20.5 million; [117]) continue to receive disproportionately more attention. Thus, it is important to ensure that adequate resources are devoted to preventing and mitigating the wildfire hazard, including but not limited to, educating the public on the dangers of carelessness with managed fires. Additional resources will be needed to combat the anticipated increased risk, especially in places where the population is expected to increase.

The results from the sensitivity analysis offer insights regarding not only the uncertainty in projections but also strategies for mitigating the future hazard. For example, these results implied that investment in improved warning and response systems, which would reduce $p(d|f_i)$, and the development of fire-retardant structure and content materials, which would reduce $d$, would seem to be wise mitigation measures. Such strategies would likely overcome or at least strongly buffer any higher-than-anticipated climate change-driven increases to the wildfire hazard, at least in Louisiana. The hope is that any reductions in the ability to acquire resources to prevent or mitigate the hazard are offset by improvements and innovations in technology to detect and combat the fires. If the risk does indeed remain roughly proportionate to the population increase, resources to combat the risk should be available, assuming that other economic and demographic factors also change proportionately. For example, the vulnerability to wildfire under the uncertain and complex conditions of response during the COVID-19 pandemic [118] would almost certainly exacerbate losses [119].

## 8. Limitations

As in any research, this work had some caveats that should be acknowledged. The lack of consideration of some natural features could limit the reliability of the findings. For example, future projections of the property damage from the hazard did not consider changes to vegetation type, fuel load, land use, or disturbance patterns, which might change the future potential destructiveness of the fires. Recent research [120] suggested that the fraction of available water (FAW) is a better predictor of large growing-season wildfires than the KBDI. FAW is calculated as the ratio of plant available water to soil water capacity. However, FAW has not yet been projected as confidently to 2050 as precipitation, and until it can be predicted better, soil moisture projections are limited to those using the KBDI.

Another set of limitations involves the population projection methodology, which ignores abrupt changes in the future, such as migrations prompted by hurricanes or other natural disasters [121], economic depression, or other extreme events. Another limitation is the assumption, necessitated by data availability, of equal population growth rates for every census block within a parish. Census-block-level population data are only available at the decennial census; interim estimates are not provided at the census block level. Therefore, we projected the population of a census block assuming that its population will grow at the same rate as the parish in which it resides. This assumption is reasonable because in most cases, the inhabited sections of a parish are small and the census blocks therein are clustered. These small, clustered areas are likely to be affected similarly by economic conditions, natural hazards, and other factors that would cause population changes. Therefore, it is likely that the growth rate is similar. In addition, population growth may not follow an exponential growth curve. This research also did not consider changes in economic and demographic characteristics by 2050, which might alter the ability of taxes to cover additional mitigation and prevention strategies. The absence of reliable demographic projections for Louisiana based on more elaborate modeling necessitates these

assumptions. As the spatial patterns of population change, one would expect the spatial patterns of human-caused ignitions and fire likelihood to shift as well. These dynamics were not represented in the current study; however, improvements and refinements in these assumptions can be directly used in the developed methodology.

Furthermore, the loss calculations were limited by many assumptions. For example, building replacement cost rather than building value could have given additional information that would have assisted in some aspects of planning, such as for setting fire insurance premiums. Likewise, the inclusion of industrial and commercial structures, crop value, and especially timber resource value would have given a much more accurate estimate of the total direct economic impact of wildfire. Finally, non-quantifiable features and assets are important in community-level wildfire mitigation planning [122] but did not appear in this property risk assessment.

**9. Summary/Conclusions**

This research developed a method for analyzing historical and future property losses to wildfire in Louisiana, a U.S. state with a relatively dense population, abundant infrastructure, and a likely increasing susceptibility to long periods without rainfall. In contrast to most work on spatial distribution of hazards, the analysis was done at the census-block level, which provided a more suitable areal unit of analysis than the parish (i.e., county) because of the fine-scale spatial variability and disparities in population, property, and in some cases, natural vulnerability to the hazard.

Wildfire is a USD 5.6 million (in 2010 USD) hazard in the state, and is projected to double by 2050 as the population grows, development along the WUI intensifies, and amplified climatic changes combine to exacerbate the risk. However, the present areas of maximum risk—west-central, east-central, and extreme northwestern and southwestern coastal Louisiana—were projected to remain the most vulnerable areas to this often-overlooked hazard by 2050.

Wildfire risk assessment can be enhanced in future research by using ignition location modeling or similar techniques to align future wildfire with shifts in population, especially as population changes align with changing vegetation and/or land use/land cover types. In addition, crop and forestry loss assessment due to wildfire could be conducted, particularly because these are of such high value in many places, including Louisiana.

Future research should also be conducted to extend a similar methodology to other hazards in other places, such as earthquakes, sinkholes, lightning, and hail. In a more general sense, improved population, economic, and demographic forecasts are needed, so that current and future risks to natural hazards, including wildfire, can be assessed more accurately. As the accuracy of climate models improves, the reliability of future projections for natural hazard risk will advance in their mission of protecting lives and property.

**Supplementary Materials:** The following are available online at https://www.mdpi.com/article/10.3390/cli10040049/s1.

**Author Contributions:** Conceptualization, R.B.M. and C.J.F.; methodology, R.B.M. and C.J.F.; software, R.B.M.; validation, R.B.M.; formal analysis, R.B.M.; investigation, R.B.M.; resources, C.J.F.; data curation, R.B.M.; writing—original draft preparation, R.B.M. and N.B.; writing—review and editing, R.V.R.; visualization, R.B.M. and C.J.F.; supervision, C.J.F. and R.V.R.; project administration, C.J.F. and R.V.R.; funding acquisition, C.J.F. and R.V.R. All authors have read and agreed to the published version of the manuscript.

**Funding:** This project resulted from the 2019 Louisiana State Hazard Mitigation Plan update, for which C.J.F. and R.V.R. received funding from FEMA, via Louisiana's Governor's Office of Homeland Security and Emergency Preparedness (GOHSEP), grant number: 2000301135. Any opinions, findings, conclusions, and recommendations expressed in this manuscript are those of the authors and do not necessarily reflect the views of FEMA or GOHSEP. Publication of this article was subsidized by the LSU Libraries Open Access Author Fund.

**Institutional Review Board Statement:** Not applicable.



**Informed Consent Statement:** Not applicable.

**Data Availability Statement:** The data presented in this study are available on request from the corresponding author.

**Acknowledgments:** The authors warmly appreciate the assistance and support of Jeffrey Giering of GOHSEP for overall project support, and David Dunaway of LSU Libraries for assistance in acquiring population data. Bret Lane of the Louisiana Department of Agriculture and Forestry provided wildfire damage data.

**Conflicts of Interest:** The authors declare no conflict of interest.

## Appendix A. Historical (1992–2015) Wildfire Burn Probability (%) by Louisiana Parish

| Parish | Point-Based | | Parishwide | |
|---|---|---|---|---|
| | Min | Max | Mean | Standard Deviation |
| Acadia | 0.01 | 1.18 | 0.16 | 0.18 |
| Allen | 1.20 | 6.22 | 3.77 | 1.28 |
| Ascension | 0.01 | 1.04 | 0.36 | 0.24 |
| Assumption | 0.00 | 0.05 | 0.01 | 0.01 |
| Avoyelles | 0.06 | 1.23 | 0.29 | 0.21 |
| Beauregard | 2.01 | 5.39 | 3.87 | 0.47 |
| Bienville | 0.82 | 1.52 | 1.17 | 0.14 |
| Bossier | 0.71 | 2.06 | 1.14 | 0.24 |
| Caddo | 0.65 | 3.20 | 1.59 | 0.56 |
| Calcasieu | 0.13 | 6.48 | 1.82 | 1.07 |
| Caldwell | 0.45 | 1.64 | 0.99 | 0.27 |
| Cameron | 0.00 | 7.70 | 1.52 | 1.61 |
| Catahoula | 0.07 | 1.24 | 0.53 | 0.31 |
| Claiborne | 1.11 | 1.56 | 1.37 | 0.11 |
| Concordia | 0.06 | 0.46 | 0.18 | 0.08 |
| De Soto | 0.61 | 1.72 | 0.83 | 0.20 |
| East Baton Rouge | 0.05 | 2.92 | 0.64 | 0.54 |
| East Carroll | 0.06 | 0.36 | 0.21 | 0.07 |
| East Feliciana | 0.15 | 3.67 | 1.72 | 0.86 |
| Evangeline | 0.25 | 5.23 | 1.73 | 1.06 |
| Franklin | 0.07 | 0.78 | 0.24 | 0.14 |
| Grant | 1.18 | 2.07 | 1.74 | 0.20 |
| Iberia | 0.00 | 0.02 | 0.00 | 0.00 |
| Iberville | 0.00 | 0.36 | 0.03 | 0.04 |
| Jackson | 0.84 | 1.22 | 1.03 | 0.08 |
| Jefferson | 0.00 | 0.93 | 0.13 | 0.18 |
| Jefferson Davis | 0.04 | 2.52 | 0.72 | 0.56 |
| Lafayette | 0.01 | 0.09 | 0.02 | 0.01 |
| Lafourche | 0.00 | 0.08 | 0.01 | 0.01 |

| Parish | Point-Based | | Parishwide | |
|---|---|---|---|---|
| | **Min** | **Max** | **Mean** | **Standard Deviation** |
| LaSalle | 0.32 | 2.01 | 1.45 | 0.37 |
| Lincoln | 0.85 | 1.33 | 1.07 | 0.12 |
| Livingston | 0.51 | 4.41 | 2.15 | 0.95 |
| Madison | 0.06 | 0.33 | 0.13 | 0.06 |
| Morehouse | 0.30 | 0.96 | 0.64 | 0.14 |
| Natchitoches | 0.82 | 1.74 | 1.13 | 0.15 |
| Orleans | 0.08 | 2.31 | 0.93 | 0.54 |
| Ouachita | 0.37 | 0.89 | 0.64 | 0.13 |
| Plaquemines | 0.00 | 0.24 | 0.03 | 0.05 |
| Pointe Coupee | 0.03 | 0.17 | 0.08 | 0.03 |
| Rapides | 0.47 | 6.13 | 2.43 | 1.46 |
| Red River | 0.65 | 1.03 | 0.83 | 0.08 |
| Richland | 0.22 | 0.67 | 0.34 | 0.07 |
| Sabine | 0.81 | 2.16 | 1.35 | 0.30 |
| St. Bernard | 0.00 | 2.00 | 0.24 | 0.21 |
| St. Charles | 0.00 | 0.49 | 0.05 | 0.06 |
| St. Helena | 2.17 | 4.75 | 3.97 | 0.48 |
| St. James | 0.01 | 0.50 | 0.10 | 0.10 |
| St. John the Baptist | 0.01 | 1.47 | 0.46 | 0.37 |
| St. Landry | 0.03 | 0.76 | 0.17 | 0.13 |
| St. Martin | 0.00 | 0.06 | 0.01 | 0.01 |
| St. Mary | 0.00 | 0.01 | 0.00 | 0.00 |
| St. Tammany | 0.92 | 5.60 | 3.15 | 0.89 |
| Tangipahoa | 1.22 | 4.76 | 3.55 | 0.96 |
| Tensas | 0.06 | 0.68 | 0.18 | 0.12 |
| Terrebonne | 0.00 | 0.02 | 0.00 | 0.00 |
| Union | 0.65 | 1.22 | 0.89 | 0.12 |
| Vermilion | 0.00 | 0.53 | 0.04 | 0.06 |
| Vernon | 1.34 | 6.11 | 3.08 | 1.12 |
| Washington | 3.19 | 6.93 | 4.14 | 0.69 |
| Webster | 0.97 | 1.54 | 1.34 | 0.10 |
| West Baton Rouge | 0.02 | 0.19 | 0.07 | 0.04 |
| West Carroll | 0.19 | 0.56 | 0.39 | 0.07 |
| West Feliciana | 0.06 | 1.39 | 0.34 | 0.26 |
| Winn | 1.10 | 1.98 | 1.48 | 0.21 |

## Appendix B. Projected Wildfire Burn Probability (%) in 2050 by Louisiana Parish

| Parish | Point-Based | | Parishwide | |
|---|---|---|---|---|
| | Min | Max | Mean | Standard Deviation |
| Acadia | 0.01 | 1.47 | 0.20 | 0.22 |
| Allen | 1.50 | 7.77 | 4.71 | 1.60 |
| Ascension | 0.02 | 1.30 | 0.45 | 0.30 |
| Assumption | 0.00 | 0.06 | 0.01 | 0.01 |
| Avoyelles | 0.08 | 1.54 | 0.36 | 0.26 |
| Beauregard | 2.51 | 6.73 | 4.83 | 0.59 |
| Bienville | 1.03 | 1.89 | 1.47 | 0.17 |
| Bossier | 0.89 | 2.57 | 1.43 | 0.31 |
| Caddo | 0.81 | 4.00 | 1.99 | 0.70 |
| Calcasieu | 0.16 | 8.10 | 2.28 | 1.33 |
| Caldwell | 0.56 | 2.05 | 1.23 | 0.33 |
| Cameron | 0.00 | 9.62 | 1.90 | 2.01 |
| Catahoula | 0.09 | 1.55 | 0.67 | 0.38 |
| Claiborne | 1.39 | 1.95 | 1.71 | 0.14 |
| Concordia | 0.08 | 0.58 | 0.22 | 0.10 |
| De Soto | 0.77 | 2.15 | 1.03 | 0.24 |
| East Baton Rouge | 0.06 | 3.65 | 0.80 | 0.67 |
| East Carroll | 0.07 | 0.45 | 0.26 | 0.09 |
| East Feliciana | 0.18 | 4.58 | 2.15 | 1.08 |
| Evangeline | 0.31 | 6.54 | 2.16 | 1.32 |
| Franklin | 0.08 | 0.98 | 0.29 | 0.17 |
| Grant | 1.47 | 2.59 | 2.17 | 0.25 |
| Iberia | 0.00 | 0.02 | 0.00 | 0.00 |
| Iberville | 0.01 | 0.45 | 0.04 | 0.05 |
| Jackson | 1.05 | 1.53 | 1.29 | 0.10 |
| Jefferson | 0.00 | 1.16 | 0.16 | 0.23 |
| Jefferson Davis | 0.05 | 3.15 | 0.89 | 0.69 |
| Lafayette | 0.01 | 0.11 | 0.03 | 0.02 |
| Lafourche | 0.00 | 0.10 | 0.01 | 0.01 |
| LaSalle | 0.40 | 2.51 | 1.81 | 0.47 |
| Lincoln | 1.06 | 1.67 | 1.34 | 0.16 |
| Livingston | 0.64 | 5.52 | 2.69 | 1.18 |
| Madison | 0.08 | 0.41 | 0.16 | 0.07 |
| Morehouse | 0.38 | 1.20 | 0.80 | 0.18 |
| Natchitoches | 1.02 | 2.18 | 1.42 | 0.18 |
| Orleans | 0.10 | 2.89 | 1.16 | 0.67 |
| Ouachita | 0.46 | 1.12 | 0.80 | 0.16 |

| Parish | Point-Based | | Parishwide | |
|---|---|---|---|---|
| | **Min** | **Max** | **Mean** | **Standard Deviation** |
| Plaquemines | 0.00 | 0.31 | 0.04 | 0.06 |
| Pointe Coupee | 0.04 | 0.22 | 0.09 | 0.03 |
| Rapides | 0.59 | 7.66 | 3.04 | 1.83 |
| Red River | 0.81 | 1.29 | 1.04 | 0.10 |
| Richland | 0.27 | 0.83 | 0.43 | 0.09 |
| Sabine | 1.02 | 2.70 | 1.69 | 0.37 |
| St. Bernard | 0.00 | 2.50 | 0.30 | 0.26 |
| St. Charles | 0.00 | 0.61 | 0.07 | 0.08 |
| St. Helena | 2.71 | 5.94 | 4.96 | 0.60 |
| St. James | 0.01 | 0.62 | 0.13 | 0.12 |
| St. John the Baptist | 0.01 | 1.84 | 0.57 | 0.47 |
| St. Landry | 0.04 | 0.95 | 0.21 | 0.16 |
| St. Martin | 0.00 | 0.07 | 0.02 | 0.01 |
| St. Mary | 0.00 | 0.02 | 0.00 | 0.00 |
| St. Tammany | 1.15 | 7.01 | 3.94 | 1.11 |
| Tangipahoa | 1.52 | 5.95 | 4.44 | 1.20 |
| Tensas | 0.08 | 0.85 | 0.23 | 0.15 |
| Terrebonne | 0.00 | 0.03 | 0.01 | 0.00 |
| Union | 0.81 | 1.52 | 1.11 | 0.15 |
| Vermilion | 0.00 | 0.66 | 0.05 | 0.08 |
| Vernon | 1.68 | 7.64 | 3.85 | 1.40 |
| Washington | 3.99 | 8.67 | 5.18 | 0.87 |
| Webster | 1.21 | 1.93 | 1.67 | 0.13 |
| West Baton Rouge | 0.03 | 0.24 | 0.08 | 0.04 |
| West Carroll | 0.24 | 0.70 | 0.49 | 0.09 |
| West Feliciana | 0.08 | 1.74 | 0.43 | 0.32 |
| Winn | 1.38 | 2.47 | 1.85 | 0.27 |

**Appendix C. Louisiana Parish Population and Population Density, Both in 2010 and Projected to 2050, and the Changes of Each**

| Parish | Population (2010) | Population (2050) | Population Change (2010–2050) | Density (per km²) (2010) | Density (per km²) (2050) | Density Change (2010–2050) (per km²) |
|---|---|---|---|---|---|---|
| Acadia | 61,773 | 67,309 | 5536 | 36.3 | 39.5 | 3.3 |
| Allen | 25,764 | 30,554 | 4790 | 13.0 | 15.4 | 2.4 |
| Ascension | 107,215 | 278,635 | 171,420 | 136.7 | 355.3 | 218.6 |
| Assumption | 23,421 | 22,875 | (546) | 24.8 | 24.2 | (0.6) |
| Avoyelles | 42,073 | 40,710 | (1363) | 18.8 | 18.2 | (0.6) |
| Beauregard | 35,654 | 45,242 | 9588 | 11.8 | 15.0 | 3.2 |

| Parish | Population (2010) | Population (2050) | Population Change (2010–2050) | Density (per km²) (2010) | Density (per km²) (2050) | Density Change (2010–2050) (per km²) |
|---|---|---|---|---|---|---|
| Bienville | 14,353 | 11,471 | (2882) | 6.7 | 5.4 | (1.4) |
| Bossier | 116,979 | 183,706 | 66,727 | 52.1 | 81.8 | 29.7 |
| Caddo | 254,969 | 238,795 | (16,174) | 105.1 | 98.4 | (6.7) |
| Calcasieu | 192,768 | 233,579 | 40,811 | 68.0 | 82.4 | 14.4 |
| Caldwell | 10,132 | 9248 | (884) | 7.2 | 6.6 | (0.6) |
| Cameron | 6839 | 5253 | (1586) | 1.4 | 1.0 | (0.3) |
| Catahoula | 10,407 | 7741 | (2666) | 5.4 | 4.0 | (1.4) |
| Claiborne | 17,195 | 15,467 | (1728) | 8.7 | 7.8 | (0.9) |
| Concordia | 20,822 | 17,145 | (3677) | 10.8 | 8.9 | (1.9) |
| De Soto | 26,656 | 28,631 | 1975 | 11.5 | 12.4 | 0.9 |
| East Baton Rouge | 440,171 | 526,522 | 86,351 | 361.4 | 432.3 | 70.9 |
| East Carroll | 7759 | 4397 | (3362) | 6.8 | 3.8 | (2.9) |
| East Feliciana | 20,267 | 20,074 | (193) | 17.2 | 17.0 | (0.2) |
| Evangeline | 33,984 | 33,924 | (60) | 19.3 | 19.3 | (0.0) |
| Franklin | 20,767 | 17,005 | (3762) | 12.6 | 10.3 | (2.3) |
| Grant | 22,309 | 29,701 | 7392 | 13.0 | 17.3 | 4.3 |
| Iberia | 73,240 | 78,687 | 5447 | 27.4 | 29.5 | 2.0 |
| Iberville | 33,387 | 33,263 | (124) | 19.7 | 19.7 | (0.1) |
| Jackson | 16,274 | 14,727 | (1547) | 10.8 | 9.8 | (1.0) |
| Jefferson | 432,552 | 409,450 | (23,102) | 260.1 | 246.2 | (13.9) |
| Jefferson Davis | 31,594 | 30,585 | (1009) | 18.5 | 17.9 | (0.6) |
| La Salle | 14,890 | 13,171 | (1719) | 8.7 | 7.7 | (1.0) |
| Lafayette | 221,578 | 361,856 | 140,278 | 317.8 | 519.0 | 201.2 |
| Lafourche | 96,318 | 112,609 | 16,291 | 25.3 | 29.6 | 4.3 |
| Lincoln | 46,735 | 54,630 | 7895 | 38.2 | 44.6 | 6.5 |
| Livingston | 128,026 | 314,726 | 186,700 | 71.5 | 175.7 | 104.3 |
| Madison | 12,093 | 8268 | (3825) | 7.2 | 4.9 | (2.3) |
| Morehouse | 27,979 | 19,510 | (8469) | 13.4 | 9.3 | (4.1) |
| Natchitoches | 39,566 | 37,548 | (2018) | 11.8 | 11.2 | (0.6) |
| Orleans | 343,829 | 310,135 | (33,694) | 379.5 | 342.3 | (37.2) |
| Ouachita | 153,720 | 167,523 | 13,803 | 93.9 | 102.4 | 8.4 |
| Plaquemines | 23,042 | 21,107 | (1935) | 3.5 | 3.2 | (0.3) |
| Pointe Coupee | 22,802 | 20,338 | (2464) | 14.9 | 13.3 | (1.6) |
| Rapides | 131,613 | 125,227 | (6386) | 37.3 | 35.5 | (1.8) |
| Red River | 9091 | 7174 | (1917) | 8.7 | 6.9 | (1.8) |
| Richland | 20,725 | 18,611 | (2114) | 14.2 | 12.7 | (1.4) |

| Parish | Population (2010) | Population (2050) | Population Change (2010–2050) | Density (per km²) (2010) | Density (per km²) (2050) | Density Change (2010–2050) (per km²) |
|---|---|---|---|---|---|---|
| Sabine | 24,233 | 22,705 | (1528) | 9.2 | 8.7 | (0.6) |
| St. Bernard | 35,897 | 59,835 | 23,938 | 6.4 | 10.7 | 4.3 |
| St. Charles | 52,780 | 74,669 | 21,889 | 51.3 | 72.6 | 21.3 |
| St. Helena | 11,203 | 11,570 | 367 | 10.6 | 10.9 | 0.3 |
| St. James | 22,102 | 21,233 | (869) | 33.1 | 31.8 | (1.3) |
| St. John the Baptist | 45,924 | 60,827 | 14,903 | 43.3 | 57.3 | 14.0 |
| St. Landry | 83,384 | 80,465 | (2919) | 34.3 | 33.1 | (1.2) |
| St. Martin | 52,160 | 68,297 | 16,137 | 24.7 | 32.3 | 7.6 |
| St. Mary | 54,650 | 41,198 | (13,452) | 18.8 | 14.2 | (4.6) |
| St. Tammany | 233,740 | 555,517 | 321,777 | 82.4 | 195.8 | 113.4 |
| Tangipahoa | 121,097 | 204,995 | 83,898 | 55.4 | 93.8 | 38.4 |
| Tensas | 5252 | 2529 | (2723) | 3.2 | 1.5 | (1.6) |
| Terrebonne | 111,860 | 129,437 | 17,577 | 20.7 | 24.0 | 3.3 |
| Union | 22,721 | 23,720 | 999 | 9.7 | 10.1 | 0.4 |
| Vermilion | 57,999 | 70,768 | 12,769 | 14.5 | 17.7 | 3.2 |
| Vernon | 52,334 | 47,403 | (4931) | 15.1 | 13.6 | (1.4) |
| Washington | 47,168 | 48,685 | 1517 | 26.9 | 27.8 | 0.9 |
| Webster | 41,207 | 35,843 | (5364) | 25.9 | 22.5 | (3.4) |
| West Baton Rouge | 23,788 | 33,301 | 9513 | 45.1 | 63.1 | 18.0 |
| West Carroll | 11,604 | 9567 | (2037) | 12.4 | 10.2 | (2.2) |
| West Feliciana | 15,625 | 19,823 | 4198 | 14.2 | 18.0 | 3.8 |
| Winn | 15,313 | 12,352 | (2961) | 6.2 | 5.0 | (1.2) |
| Louisiana | 4,533,372 | 5,661,868 | 1,128,496 | 33.4 | 41.7 | 8.3 |

**Appendix D. Historical (1992–2015) Annual Average and 2050-Projected Property Loss, per Capita Property Loss, and per Building Property Loss by Louisiana Parish (2010 USD)**

| Parish | Annual Property Loss | | Annual Per Capita Property Loss | | Annual Per Building Property Loss | |
|---|---|---|---|---|---|---|
| | Historical (1992–2015) | Projected (2050) | Historical (1992–2015) | Projected (2050) | Historical (1992–2015) | Projected (2050) |
| Acadia | 5834 | 7939 | 0.09 | 0.12 | 0.23 | 0.29 |
| Allen | 70,484 | 104,389 | 2.74 | 3.42 | 7.24 | 9.05 |
| Ascension | 79,001 | 256,591 | 0.74 | 0.92 | 1.94 | 2.42 |
| Assumption | 166 | 203 | 0.01 | 0.01 | 0.02 | 0.02 |
| Avoyelles | 13,541 | 16,395 | 0.32 | 0.40 | 0.75 | 0.94 |
| Beauregard | 132,231 | 209,560 | 3.71 | 4.63 | 8.79 | 10.99 |

| Parish | Annual Property Loss | | Annual Per Capita Property Loss | | Annual Per Building Property Loss | |
|---|---|---|---|---|---|---|
| | Historical (1992–2015) | Projected (2050) | Historical (1992–2015) | Projected (2050) | Historical (1992–2015) | Projected (2050) |
| Bienville | 14,537 | 14,661 | 1.01 | 1.28 | 1.88 | 2.36 |
| Bossier | 151,806 | 297,915 | 1.30 | 1.62 | 3.08 | 3.84 |
| Caddo | 355,593 | 416,460 | 1.39 | 1.74 | 3.17 | 3.97 |
| Calcasieu | 262,914 | 398,089 | 1.36 | 1.70 | 3.20 | 4.00 |
| Caldwell | 9166 | 10,486 | 0.90 | 1.13 | 1.84 | 2.29 |
| Cameron | 11,858 | 11,474 | 1.73 | 2.18 | 3.30 | 4.10 |
| Catahoula | 4405 | 4104 | 0.42 | 0.53 | 0.90 | 1.13 |
| Claiborne | 18,779 | 21,171 | 1.09 | 1.37 | 2.42 | 3.02 |
| Concordia | 4071 | 4208 | 0.20 | 0.25 | 0.43 | 0.54 |
| De Soto | 22,093 | 29,624 | 0.83 | 1.03 | 1.80 | 2.25 |
| East Baton Rouge | 318,443 | 476,118 | 0.72 | 0.90 | 1.70 | 2.12 |
| East Carroll | 774 | 555 | 0.10 | 0.13 | 0.27 | 0.33 |
| East Feliciana | 33,236 | 41,186 | 1.64 | 2.05 | 4.15 | 5.19 |
| Evangeline | 36,540 | 45,596 | 1.08 | 1.34 | 2.49 | 3.11 |
| Franklin | 3602 | 3702 | 0.17 | 0.22 | 0.40 | 0.50 |
| Grant | 33,168 | 55,078 | 1.49 | 1.85 | 3.73 | 4.67 |
| Iberia | 273 | 367 | 0.00 | 0.00 | 0.01 | 0.01 |
| Iberville | 1412 | 1759 | 0.04 | 0.05 | 0.11 | 0.14 |
| Jackson | 18,581 | 21,103 | 1.14 | 1.43 | 2.42 | 3.02 |
| Jefferson | 103,818 | 122,841 | 0.24 | 0.30 | 0.55 | 0.69 |
| Jefferson Davis | 12,467 | 15,121 | 0.39 | 0.49 | 0.94 | 1.17 |
| La Salle | 19,561 | 21,701 | 1.31 | 1.65 | 2.98 | 3.72 |
| Lafayette | 7293 | 14,887 | 0.03 | 0.04 | 0.08 | 0.10 |
| Lafourche | 397 | 580 | 0.00 | 0.01 | 0.01 | 0.01 |
| Lincoln | 54,301 | 79,338 | 1.16 | 1.45 | 2.79 | 3.49 |
| Livingston | 321,625 | 987,898 | 2.51 | 3.14 | 6.41 | 8.01 |
| Madison | 831 | 717 | 0.07 | 0.09 | 0.17 | 0.22 |
| Morehouse | 15,877 | 13,870 | 0.57 | 0.71 | 1.28 | 1.60 |
| Natchitoches | 47,276 | 56,097 | 1.19 | 1.49 | 2.54 | 3.18 |
| Orleans | 226,704 | 255,825 | 0.66 | 0.82 | 1.19 | 1.49 |
| Ouachita | 114,107 | 155,416 | 0.74 | 0.93 | 1.77 | 2.21 |
| Plaquemines | 3075 | 3524 | 0.13 | 0.17 | 0.32 | 0.40 |
| Pointe Coupee | 2493 | 2779 | 0.11 | 0.14 | 0.22 | 0.28 |
| Rapides | 267,425 | 318,269 | 2.03 | 2.54 | 4.80 | 6.00 |
| Red River | 6534 | 6460 | 0.72 | 0.90 | 1.58 | 1.98 |
| Richland | 4960 | 5580 | 0.24 | 0.30 | 0.58 | 0.72 |

| Parish | Annual Property Loss | | Annual Per Capita Property Loss | | Annual Per Building Property Loss | |
|---|---|---|---|---|---|---|
| | Historical (1992–2015) | Projected (2050) | Historical (1992–2015) | Projected (2050) | Historical (1992–2015) | Projected (2050) |
| Sabine | 42,303 | 49,748 | 1.75 | 2.19 | 2.99 | 3.75 |
| St. Bernard | 12,487 | 25,996 | 0.35 | 0.43 | 0.74 | 0.93 |
| St. Charles | 1623 | 2869 | 0.03 | 0.04 | 0.08 | 0.10 |
| St. Helena | 49,864 | 64,279 | 4.45 | 5.56 | 9.68 | 12.10 |
| St. James | 2187 | 2627 | 0.10 | 0.12 | 0.26 | 0.32 |
| St. John the Baptist | 9400 | 15,556 | 0.20 | 0.26 | 0.54 | 0.67 |
| St. Landry | 13,271 | 16,020 | 0.16 | 0.20 | 0.37 | 0.46 |
| St. Martin | 994 | 1625 | 0.02 | 0.02 | 0.05 | 0.06 |
| St. Mary | 44 | 41 | 0.00 | 0.00 | 0.00 | 0.00 |
| St. Tammany | 1,560,580 | 4,633,439 | 6.68 | 8.34 | 16.36 | 20.45 |
| Tangipahoa | 630,169 | 1,332,887 | 5.20 | 6.50 | 12.58 | 15.73 |
| Tensas | 1809 | 1109 | 0.34 | 0.44 | 0.54 | 0.67 |
| Terrebonne | 117 | 169 | 0.00 | 0.00 | 0.00 | 0.00 |
| Union | 21,337 | 27,811 | 0.94 | 1.17 | 1.88 | 2.35 |
| Vermilion | 650 | 988 | 0.01 | 0.01 | 0.03 | 0.03 |
| Vernon | 117,933 | 133,743 | 2.25 | 2.82 | 5.50 | 6.88 |
| Washington | 198,893 | 256,359 | 4.22 | 5.27 | 9.45 | 11.82 |
| Webster | 51,412 | 55,964 | 1.25 | 1.56 | 2.66 | 3.32 |
| West Baton Rouge | 2469 | 4319 | 0.10 | 0.13 | 0.26 | 0.33 |
| West Carroll | 3879 | 4006 | 0.33 | 0.42 | 0.77 | 0.96 |
| West Feliciana | 7375 | 11,668 | 0.47 | 0.59 | 1.45 | 1.82 |
| Winn | 16,342 | 16,634 | 1.07 | 1.35 | 2.26 | 2.82 |
| Louisiana | 5,556,389 | 11,167,496 | 1.23 | 1.97 | 2.83 | 4.63 |

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
