# Peer review of "Estimating Future Residential Property Risk Associated with Wildfires in Louisiana, U.S.A."

_climate, doi:10.3390/cli10040049_

Round 1
Reviewer 1 Report
The manuscript is well written. However I feel that the introduction section needs a bit more well structured with wildfire issues in the USA, and then for the State of Louisiana.
I see that literature review is condensed with the introduction section. For general readers it would be helpful to have the literate and introduction part separated with subtitle.
Also in the results section: I see the authors reported different regions as "The west-central, east-central, and extreme northwestern 376 and southwestern parts of the state have the highest burn probability for wildfire (Figure 377 1A)." It would be wise to include the historic wildfire incidents by region in the State of Louisiana if possible.
The figure 2." Population density by census block: (A) 2010, and (B) change in population density from 408 2010 to 2050"- is poorly presented in terms of symbology and color choice. A cartographically sound map with better color balance should be included.
Author Response
The manuscript is well written. However I feel that the introduction section needs a bit more well-structured with wildfire issues in the USA, and then for the State of Louisiana.
Reply: In our revised submission, the separation of the (former) Introduction section into four sections, as described in our response to the Reviewer’s next comment below, builds an improved structure on wildfire issues. There isn’t much in the literature about Louisiana wildfires, but we include the literature that does exist on Louisiana wildfires in the “Study Area” section. We structured the (new) Background section around papers that emphasize the wildfire impact on property value, and then we proceed to discuss literature that emphasizes population changes and mobility, and then we describe the more sophisticated models that include uncertainty. We end the (new) Background section with a specific look at FSim, which is the fire model used to assess the losses and risk in our research. Then we include a short (new) section entitled, “Temporal Trends in Wildfire Occurrence in the U.S.A.” which provides a smooth transition to the “Study Area” section. We appreciate this suggestion from Reviewer #1.
I see that literature review is condensed with the introduction section. For general readers it would be helpful to have the literate and introduction part separated with subtitle.
Reply: As described above, in our revised submission we divided the (former) Introduction section into four sections. A now-shorter Introduction section takes the reader to the objectives of the paper, focused on property risk assessment to wildfire, much earlier in the manuscript. Then, a Background section “digs deeper” into the current literature on property risk assessment, progressing chronologically within three types of property risk research. Then, we follow with a brief section on the temporal trends in wildfire frequency in the U.S.A., in order to lead into our Study Area section (which had existed previously). We are proud of these changes, as we feel that they improve the logic, flow, and readability of the paper. We are grateful to Reviewer #1 for forcing us to think more closely about this part of the paper. These changes necessitated many changes to the text, so that we could disentangle the text into this new framework. In our revised submission, we show all of the changes made in “Track Changes” mode.
Also in the results section: I see the authors reported different regions as "The west-central, east-central, and extreme northwestern and southwestern parts of the state have the highest burn probability for wildfire (Figure 1A)." It would be wise to include the historic wildfire incidents by region in the State of Louisiana if possible.
Reply: In our revised submission, we include a small table within Figure 1A that lists historical wildfire incidents (1992─2015) for the top 12 parishes in the state, with the ranking placed inside the corresponding parish’s boundary. This addition also helps the reader to see the location of parishes mentioned in the text, without the need for adding an additional map or table. We thank Reviewer #1 for this helpful suggestion.
The figure 2." Population density by census block: (A) 2010, and (B) change in population density from 2010 to 2050"- is poorly presented in terms of symbology and color choice. A cartographically sound map with better color balance should be included.
Reply: We agree that Figure 2 was confusing in its previous form. We had shown population density, which of course creates complications when census blocks are of different sizes. While every strategy for mapping population has its drawbacks, Reviewer #1’s comment convinced us to map population and population change in Figures 2A and 2B, respectively, rather than population density and population density change. In our updated version, we use shades of gray to depict population (in Figure 2A), so that the reader can see the relative sizes of Louisiana’s cities, and the settlement patterns (particularly along the rivers/bayous and away from the low-lying coast). The reader is also left with a sense that so much of the state is unpopulated (white). We feel that this version of Figure 2A offers a great improvement as the first population map in the paper. Figure 2B was also changed substantially. The (former) shades of gray/black for population gains was illogical, as Reviewer #1 noted. In our new Figure 2B, in addition to the change from showing population density change to population change, we use red to depict population loss and shades of green to depict population gains (2010─2050 in each case), by census block. We also enlarged the legend in Figure 2 and in the other maps in the paper.

Reviewer 2 Report
A well written manuscript detailing historical aspects of wildfire and utilising sensitivity tests to determine validity of results. It would add value to advance current knowledge-base, especially the nexus between global warming and wildfire risk. Louisiana was used as a case. However there are few issues that must be considered to improve its readership. They are:
- The title should be ‘Estimating future residential property risk associated with wildfires in Louisiana, USA’ – changing to this could reflect the content properly
- Abstract: line 13: ‘examined’. Line 25: Overall, results suggested …
- Data: do not talk about your objectives here, this section requires a bit editing
Author Response
A well written manuscript detailing historical aspects of wildfire and utilising sensitivity tests to determine validity of results. It would add value to advance current knowledge-base, especially the nexus between global warming and wildfire risk. Louisiana was used as a case. However there are few issues that must be considered to improve its readership. They are:
- The title should be ‘Estimating future residential property risk associated with wildfires in Louisiana, USA’ – changing to this could reflect the content properly
Reply: Changed accordingly.
- Abstract: line 13: ‘examined’. Line 25: Overall, results suggested …
Reply: Both lines are changed accordingly.
- Data: do not talk about your objectives here, this section requires a bit editing
Reply: In our revised submission, we removed discussion of the objectives in the Data section; instead the objectives are provided earlier in the paper in our (now-shortened, in response to Reviewer #1’s comments) “Introduction” section.

Reviewer 3 Report
- General comments
The manuscript is well written and from the standpoint of English, it is very good. The manuscript examines historical and future wildfire property risk at the census-block level in Luisiana (USA), which characterized by dense population and wet climate. Firstly, there are no field measurements, at least just for an accuracy assessment. Also, the fires near Louisiana are small (<120 hectares) to talk about wildfire disasters. The manuscript does not consider important factors as vegetation, land use changes etc. Also, you mention too much the property loss with monetary costs, for example in the abstract, which from my point of view is not suitable for a scientific doc. In “conclusions” you support that you present a method for analyzing historical/future property losses to wildfire considering a different approach, but the quality of presentation is low and not clear at all.
- Some more comments
Some of the figures (eg Figure 3, Figure 4) in the text are difficult to read them and need reform (too small numbers and depiction). Also, from my point of view the text in 2.2 chapter could be improved avoiding mention tools from the software. Additionally, I have to point out that “Discussion” needs to be expanded, compared with the previous chapters, and do not comment only the limitations of the paper. Also, the references need a re-working.
- Some detailed comments line by line:
Line 13: why you use current?
Line 13: In my opinion, wildfire is not an underestimated hazard
Line 54: References missing here.
Line 54: you mention the “health” impact of the fire, but you do not considered it at all.
Line 157-165: I believe that is better to summarize here the goal of the manuscript
Author Response
The manuscript is well written and from the standpoint of English, it is very good. The manuscript examines historical and future wildfire property risk at the census-block level in Louisiana (USA), which characterized by dense population and wet climate. Firstly, there are no field measurements, at least just for an accuracy assessment.
Reply: The last sentence in the “Data” (now 5.1) subsection specifies that Louisiana Department of Agriculture & Forestry (LDAF) detailed fire summary data for Louisiana (2007–2017; [96]) serve as a baseline for future property loss due to wildfire. These data are collected as primary data in the field and is now uploaded as “supplementary documents” along with the revised version.
Also, the fires near Louisiana are small (<120 hectares) to talk about wildfire disasters. The manuscript does not consider important factors as vegetation, land use changes etc.
Reply: The reviewer is only partially correct. FSim, which is described in the context of previous research in the (new) “Background: Wildfire-related Property Impacts” (Section 2) and in the context of our own research in “Assessing historical wildfire burn probability” (Subsection 5.2), does indeed consider vegetation when calculating burn probability. Thus, our use of data from Short (2017) and Short et al. (2020) for assessing current wildfire loss implicitly includes these factors. In our original version of the manuscript, we were not clear in this regard, so in our revised manuscript (Lines 121─122; last sentence of 2. “Background: Wildfire-related Property Impacts” section and Lines 158─159; third sentence of 5.1 “Data” section), we clarify that vegetation and other land use are indeed included in the current wildfire simulations.
However, in assessing the future wildfire risk, we did not take into account any changes in vegetation or other land use. To do so would require land use change projections that are unavailable to us, especially at a scale that would be congruent with the rest of the analysis. This limitation was acknowledged in the earlier version of the manuscript and it now appears in the first paragraph of the (new) Limitations section and in the first sentence of the penultimate paragraph of the “Summary/Conclusions” section. Our revised version also includes the following clarifying insertion in the third sentence of the first paragraph of the new “Limitations” section: “For example, future projections of the property damage from the hazard do this work does not consider changes to vegetation type...”
Also, you mention too much the property loss with monetary costs, for example in the abstract, which from my point of view is not suitable for a scientific doc.
Reply: We aren’t sure what the reviewer is suggesting in this comment. If Reviewer #3 is suggesting that monetary figures are not appropriate to discuss in a scientific document, we respectfully disagree. Careful standardization of monetary values to a specified year for the value of the dollar is standard in much scientific research. In fact, the public demands that we as scientists help them to ascertain “what it all means,” and very often the most appropriate way to assign importance to a problem is to attach a dollar value to it. We believe that this research, like so much existing natural hazards risk and impacts research, falls into this category.
If, however, Reviewer #3 is expressing concern that the precision with which we express the results of the monetary losses, we agree. We changed Lines 20 and 23 of the Abstract to report rounded numbers. We also added a new first sentence to the second paragraph of the “Summary/Conclusions” section that uses rounded numbers. We feel that in the Results section it is appropriate to present the exact numbers that we found from the work.
In “conclusions” you support that you present a method for analyzing historical/future property losses to wildfire considering a different approach, but the quality of presentation is low and not clear at all.
Reply: We appreciate the observation of the reviewer and we agree that we had a poor flow of ideas in the second paragraph of the “Summary/Conclusions” section. We also included a few clarifying words/phrases in the other paragraphs in our revised submission.
- Some more comments
Some of the figures (eg Figure 3, Figure 4) in the text are difficult to read them and need reform (too small numbers and depiction).
Reply: To enhance legibility, we enlarged the legends in each of the figures, in addition to the more substantive additions to Figure 2, as we described previously. The printed version of the figures will be at higher resolution than it appears when embedded in the Word files. Upon request from the Editor and/or production staff, we can provide any/all figures at higher resolution than included in this revised submission.
Also, from my point of view the text in 2.2 chapter could be improved avoiding mention tools from the software.
Reply: We removed the mention of the tools from the software, in both places in which these were mentioned in the earlier submission.
Additionally, I have to point out that “Discussion” needs to be expanded, compared with the previous chapters, and do not comment only the limitations of the paper.
Reply: In our revised submission, we expanded the “Discussion” section by providing better context for our results – through comparison of the magnitude of the wildfire hazard to other hazards in Louisiana. We also separated the “Discussion” from the “Limitations” by moving “Limitations” to its own section.
Also, the references need a re-working.
Reply: In our revised version, we converted the referencing style from the former approach with the author name and year to a subscripted numerical approach, as required in Climate.
- Some detailed comments line by line:
Line 13: why you use current?
Reply: We changed the word from “current” to “historical.”
Line 13: In my opinion, wildfire is not an underestimated hazard
Reply: We inserted “in some areas” in this sentence after “understudied,” to specify that the wildfire hazard needs more attention in certain places.
Line 54: References missing here.
Reply: This sentence comes from our own synthesis of the literature, which is referenced in the rest of the paragraph. This is just a general topic sentence for the paragraph.
Line 54: you mention the “health” impact of the fire, but you do not considered it at all.
Reply: We agree that we do not consider the health impacts, nor do we consider the environmental impacts, of wildfire. The first sentence of the paragraph in question (now the third paragraph of the “Introduction” section) outlines the three major types of impacts, and to zero in on where our research fits into this framework, we begin the next paragraph by saying, “This research presents a census-block-level property risk assessment for wildfire in Louisiana, U.S.A., in contrast to…” Future work must be done to properly consider health and environmental impacts of wildfire in Louisiana; our paper couldn’t possibly have treated these important considerations adequately in addition to examining the property risks.
Line 157–165: I believe that is better to summarize here the goal of the manuscript
Reply: We particularly appreciate this suggestion. The objectives of the manuscript had already been summarized in Lines 130–143 of the original submission. In the revised manuscript, we tell the reader the objectives earlier in the paper (i.e., in Lines 58–72), at the end of the Introduction section, which is where most readers and reviewers (including Reviewer 3) generally expects to find them. Also, as described in our earlier responses to reviewer comments, our revised submission includes a new Section 4 (“Study Area”).

Round 2
Reviewer 3 Report
I found the manuscript now more accurate and clearer.
Some corrections for the manuscript:
- Supplementary: I have to point out that the first plot is not very useful. Also, you can add titles in the axes.
- Line 208 (new): please delete the double space
- Chapter 5.2: I think is better now or you can add for example “To extract the small fire probabilities, the nationwide fire occurrence ………….. fires larger than 300 acres are removed using GIS applications”.
- References: in my opinion this reference “Mostafiz, R.B.; Bushra, N.; Rohli, R.V.; Friedland, C.J. Present vs. future losses from a 100-year flood: A case study of Grand 654 Isle, Louisiana. In American Geophysical Union Conference, New Orleans, USA, 13/12/2021.” should be excluded, because firstly is the same with No1 and secondly are many self-citations.
Author Response
Some corrections for the manuscript:
- Supplementary: I have to point out that the first plot is not very useful. Also, you can add titles in the axes.
Reply: Changed accordingly.
- Line 208 (new): please delete the double space
Reply: Changed accordingly.
- Chapter 5.2: I think is better now or you can add for example “To extract the small fire probabilities, the nationwide fire occurrence ………….. fires larger than 300 acres are removed using GIS applications”.
Reply: Changed accordingly.
- References: in my opinion this reference “Mostafiz, R.B.; Bushra, N.; Rohli, R.V.; Friedland, C.J. Present vs. future losses from a 100-year flood: A case study of Grand 654 Isle, Louisiana. In American Geophysical Union Conference, New Orleans, USA, 13/12/2021.” should be excluded, because firstly is the same with No1 and secondly are many self-citations.
Reply: Changed accordingly.
